# Improved Depth Estimation of Bayesian Neural Networks

**Bart van Erp**[1,2]          **Bert de Vries**[1,2,3]

[1] Lazy Dynamics    [2] Eindhoven University of Technology    [3] GN Hearing

Eindhoven, The Netherlands

{b.v.erp, bert.de.vries}@tue.nl

## Abstract

This paper proposes improvements over earlier work by Nazareth and Blei [1] for estimating the depth of Bayesian neural networks. Here, we propose a discrete truncated normal distribution over the network depth to independently learn its mean and variance. Posterior distributions are inferred by minimizing the variational free energy, which balances the model complexity and accuracy. Our method improves test accuracy on the spiral data set and reduces the variance in posterior depth estimates.

## 1   Introduction

Determining the optimal neural network architecture for a given problem is a challenging task, typically involving manual design iterations or automated grid searches. Both approaches are time-consuming and resource-intensive. A critical aspect of this process is balancing the model's complexity to prevent overfitting while ensuring high accuracy.

The seminal work of Nazareth and Blei [1] introduced a variational inference scheme to network depth estimation. By treating the layer depth of the model as a latent variable, they can infer its posterior distribution. Importantly, their variational free energy provided an excellent objective for balancing the model complexity against the model accuracy.

Although the approach presented in [1] offers a refreshing perspective, some areas could be improved. For instance, using a truncated Poisson distribution for layer depth results in the mean and variance being approximately equal, which can lead to significant uncertainty in determining the appropriate number of layers, especially for networks of increasing depth and complexity. Moreover, although the methodology in [1] is based on variational principles, certain simplifying assumptions undermine the probabilistic nature of their model. Specifically, the first-order linearization approximation over expectations neglects uncertainties over the parameters.

This paper focuses exclusively on Bayesian neural networks and builds on the work by [1], addressing the aforementioned areas of improvement. Specifically, we make the following contributions:

- We propose a discrete truncated normal distribution over the number of hidden layers of a Bayesian neural network, enabling variance reduction in the posterior estimates of the appropriate number of layers;
- Parameter estimation and structure learning are jointly performed by minimization of the variational free energy, explicitly taking the uncertainties over variables into account.

In Section 2 the probabilistic model is specified, after which the inference procedure is elaborated in Section 3. Section 4 discusses the results obtained, and Section 5 concludes the paper.

Workshop on Bayesian Decision-making and Uncertainty, 38th Conference on Neural Information Processing Systems (NeurIPS 2024).

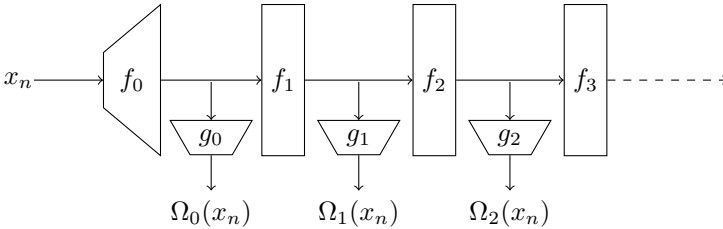

Figure 1: Visualization of the non-linearity $\Omega_L$ in (2). Deeper models reuse parts of shallower models.

## 2 Model specification

Let $\mathcal{D} = \{(x_n, y_n)\}_{n=1}^N$ be a dataset of $N$ labeled observations. We define the likelihood function of a Bayesian neural network as

$$p(y_n \,|\, x_n, \theta, L) = \mathcal{N}(y_n \,|\, \Omega_L(x_n), \Sigma), \tag{1a}$$

$$p(y_n \,|\, x_n, \theta, L) = \mathrm{Cat}(y_n \,|\, \sigma(\Omega_L(x_n))), \tag{1b}$$

for regression and classification, respectively. $\mathcal{N}(\cdot \,|\, \mu, \Sigma)$ represents a normal distribution with mean $\mu$ and covariance $\Sigma$ and $\mathrm{Cat}(\cdot \,|\, p)$ is a categorical distribution with event probabilities $p$, with $\sigma(\cdot)$ denoting the softmax function. The underlying non-linearity $\Omega_L$ is parameterized by parameters $\theta$, is visualized in Figure 1 and is defined as the composition

$$\Omega_L = g_L \circ f_L \circ f_{L-1} \circ \cdots \circ f_1 \circ f_0, \tag{2}$$

with input transformation $f_0$, latent transformations $\{f_l\}_{l=1}^L$ and output transformations $\{g_l\}_{l=0}^L$.

We treat the model depth $L \in \mathbb{N}_0$ as an unknown variable. Therefore, a suitable discrete prior must be selected, with limited support and enabling efficient inference. The truncated Poisson distribution proposed in [1] has a variance and support that grows in network depth, preventing it from converging to a single value for the depth.

Alternative discrete distributions suffer from similar problems, such as the negative binomial distribution whose variance is always larger or equal to its mean. Others do not have continuous parameters, such as the hypergeometric distribution with integer parameters. For the categorical distributions used in [2], the support needs to be bounded. The generalized Poisson distribution [3] enables situations where its mean exceeds its variance, however, in those situations the distribution quickly becomes ill-defined [4]. Furthermore, the Conway-Maxwell-Poisson distribution does not require closed-form expressions for its normalization constant [5, 6].

Here, we propose to use a discrete truncated normal distribution, whose mean and variance are decoupled, which enables us to model both over- and under-dispersed distributions. Let $\mathcal{N}_{\geq 0}(x \,|\, \mu, \sigma^2) \overset{\triangle}{\propto} \mathcal{N}(x \,|\, \mu, \sigma^2) \mathbb{1}[x \,|\, x \geq 0]$ denote a normal distribution truncated to the positive real line. Based on this truncated normal distribution, we define the prior over $L$ as its discrete counterpart

$$p(L) = \int_L^{L+1} \mathcal{N}_{\geq 0}(l \,|\, \mu_L, \sigma_L^2) \,\mathrm{d}l \qquad \text{for } L \in \mathbb{N}_0. \tag{3}$$

We intentionally do not choose a discrete Gamma distribution [7] here, despite its positive domain, because computing derivatives to the shape parameter after truncation is difficult due to the presence of the lower incomplete gamma function in its cumulative density function.

To complete the model specification, the prior over the parameters is chosen to fully factorize as

$$p(\theta \,|\, L) = \prod_{\vartheta_{g_L} \in \theta_{g_L}} \mathcal{N}(\vartheta_{g_L} \,|\, \mu_\vartheta, \sigma_\vartheta^2) \prod_{l=0}^L \prod_{\vartheta_{f_l} \in \theta_{f_l}} \mathcal{N}(\vartheta_{f_l} \,|\, \mu_\vartheta, \sigma_\vartheta^2), \tag{4}$$

where an explicit distinction in made between the parameters in the input and hidden layers $\{\theta_{f_l}\}_{l=0}^L$, which are shared amongst different model depths, and in the depth-specific output layers $\{\theta_{g_l}\}_{l=0}^L$.

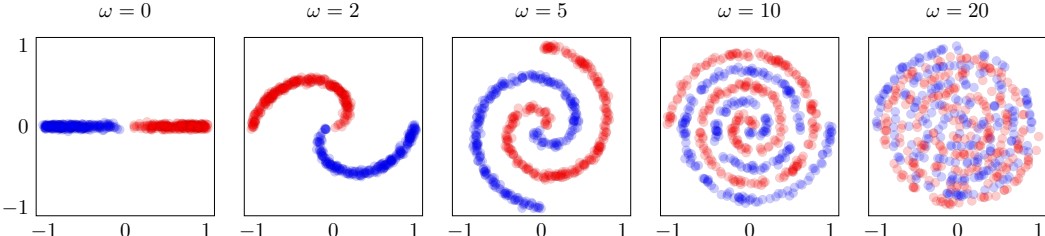

Figure 2: Spiral datasets for different rotation speeds $\omega$, generated according to Appendix A.1.

With the model specified, the next step involves specifying the variational posterior distribution. We factorize the variational posterior distribution as

$$q(\theta, L) = q(L) \prod_{\vartheta_{g_L} \in \theta_{g_L}} \mathcal{N}(\vartheta_{g_L} \,|\, \hat{\mu}_\vartheta, \hat{\sigma}_\vartheta^2) \prod_{l=0}^{L} \prod_{\vartheta_{f_l} \in \theta_{f_l}} \mathcal{N}(\vartheta_{f_l} \,|\, \hat{\mu}_\vartheta, \hat{\sigma}_\vartheta^2). \tag{5}$$

To retain tractability, we further truncate the variational posterior distribution over $L$ to its lower and upper quantiles defined by $p_l$, $p_u$ to ensure a limited support by defining

$$\mathcal{N}_{\geq 0}^{[p_l, p_u]}(x \,|\, \mu, \sigma^2) \overset{\triangle}{\propto} \mathcal{N}_{\geq 0}(x \,|\, \mu, \sigma^2) \mathbb{1}\left[ x \,\Big|\, p_l \leq \int_0^x \mathcal{N}_{\geq 0}(z \,|\, \mu, \sigma^2)\,\mathrm{d}z \leq p_u \right]. \tag{6}$$

Using this expression the variational posterior distribution over the network depth is formulated as

$$q(L) = \int_L^{L+1} \mathcal{N}_{\geq 0}^{[p_l, p_u]}(l \,|\, \hat{\mu}_L, \hat{\sigma}_L^2)\,\mathrm{d}l \qquad \text{for } L \in \mathbb{N}_0. \tag{7}$$

where the $\hat{\ }$ accent identifies the variational parameters in (6) and (7).

## 3   Probabilistic inference

Estimation of the variational posterior distributions, which encompasses both parameter estimation and structure learning, is achieved by minimization of the variational free energy

$$\begin{aligned}
\mathrm{F}[p, q] &= \mathbb{E}_{q(L,\theta)}\left[ \ln \frac{p(y, \theta, L \,|\, x)}{q(\theta, L)} \right], \\
&= \mathbb{E}_{q(L)}\left[ \ln \frac{q(L)}{p(L)} + \mathbb{E}_{q(\theta \,|\, L)}\left[ \ln \frac{q(\theta \,|\, L)}{p(\theta \,|\, L)} + \sum_{n=1}^{N} \ln p(y_n \,|\, x_n, \theta, L) \right] \right],
\end{aligned} \tag{8}$$

where the expectation over parameters can be further decomposed as

$$\mathbb{E}_{q(\theta \,|\, L)}\left[ \ln \frac{q(\theta \,|\, L)}{p(\theta \,|\, L)} \right] = \sum_{\vartheta_{g_L} \in \theta_{g_L}} \mathrm{KL}\left[ q(\vartheta_{g_L}) \| p(\vartheta_{g_L}) \right] + \sum_{l=0}^{L} \sum_{\vartheta_{f_l} \in \theta_{f_l}} \mathrm{KL}\left[ q(\vartheta f_l) \| p(\vartheta_{f_l}) \right]. \tag{9}$$

Although the expectation over the network depth seems computationally involved, the limited support as a result of the truncation in (7) reduces this operation to a finite summation as $\mathbb{E}_{q(L)}[f(\cdot \,|\, L)] = \sum_{l \in \mathrm{supp}\{q(L)\}} q(l) f(\cdot \,|\, l)$. Furthermore, since hidden layers are reused in networks of varying depth as illustrated in Figure 1, most computations can be reused in computing the expected log-evidence.

## 4   Experiments

All experiments[1] have been implemented in Julia [8] to explore its excellent metaprogramming capabilities as required by the dynamic nature of the unbounded models. We closely follow the experimental design of [1] and generate a train, validation and test set of 1024 samples each of

---

[1]All experiments are publicly available at `https://github.com/biaslab/DepthEstimationBNN`.

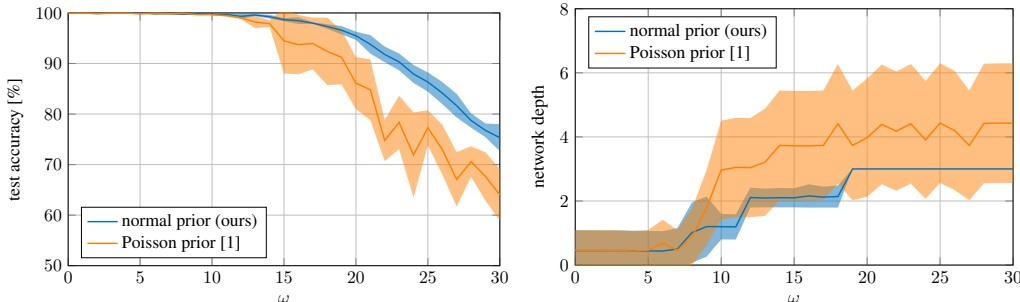

Figure 3: (Left) Test accuracy on the spiral classification task for varying rotation speeds $\omega$. Solid lines represent the average accuracy over five independent runs, with shaded areas indicating one standard deviation ($\pm\sigma$). The discrete truncated normal distribution shows accuracy improvements across all rotational speeds compared to the Poisson-based model in [1]. (Right) Means and standard deviations of the posterior distributions over network depth, shown for the first run, with similar trends across other runs. As expected, the variance of the Poisson-based model increases at larger depths, while the normal distribution converges to a single depth.

the spiral dataset [1, 9] for binary classification as described in Appendix A.1. This dataset is parameterized by a rotation speed $\omega$, which captures the difficulty of the dataset as shown in Figure 2.

The input layer $f_0 : \mathbb{R}^2 \to \mathbb{R}^{32}$ and latent layers $f_l : \mathbb{R}^{32} \to \mathbb{R}^{32} \, \forall l \geq 1$ each consist of a linear transformation followed by a LeakyReLU [10]. The output layers only involve a linear transformation $g_l : \mathbb{R}^{32} \to \mathbb{R}^2 \, \forall l \geq 0$, where the non-linearity appears in (1b). We compare our approach to [1] which uses a $\mathrm{Poisson}(0.5)$ prior, where the variational posterior distribution is initialized by the $\mathrm{Poisson}(1.0)$ distribution, truncated to the $0.95$-quantile. We select a similarly shaped normal distribution ($\mu_L = 0, \hat{\mu}_L = 0, \sigma_L = 1.15$ and $\hat{\sigma}_L = 1.8$), whose truncation is defined by $p_l = 0.025$ and $p_u = 0.975$. Appendix A.2 shows the resemblance between these priors.

We jointly learn the parameters of the probabilistic model and its variational posterior through stochastic variational inference [11] by minimizing the variational free energy in (8) using the Adam optimizer [12] until convergence. Appendix A.3 specifies the hyperparameter settings. Inference in the model is performed using Bayes-by-backprop [13] with local reparameterization [14]. The model that achieves the lowest variational free energy on the validation set is saved and evaluated on the test set by forming predictions according to

$$p(y^\star \,|\, x^\star) \approx \mathbb{E}_{q(\theta,L)} \left[ p(y^\star \,|\, x^\star, \theta, L) \right]. \tag{10}$$

Figure 3 shows the achieved predictive accuracy on the test set and the inferred posterior distributions over the model depth. From this we conclude that the discrete truncated normal distribution outperforms the Poisson distribution on the spiral classification task. The normal-based model achieves a higher accuracy, which becomes increasingly significant when the complexity of the data increases. Furthermore, as expected, the posterior distribution over the model depth in the normal-based model has a reduced variance in comparison to the Poisson-based model, as its mean and variance are naturally decoupled during training. In practice this leads to computational savings when making predictions using (10) as the narrow support of $q(L)$ requires less output layers $g_l$ to be active.

## 5  Discussion and conclusion

This paper introduces a discrete truncated normal distribution for modeling the depth of a Bayesian neural network and demonstrates how to infer its posterior distribution through the minimization of variational free energy. Compared to methods using a Poisson prior [1], our approach results in reduced variance in posterior estimates and improved test accuracy on the spiral classification task.

The results presented in this paper show promising improvements in estimating the depth of Bayesian neural networks. However, additional experiments are required involving more complex models and tasks. Network width estimation and parameter pruning [13, 15–17] offer valuable opportunities for further expanding the methodology presented here.

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

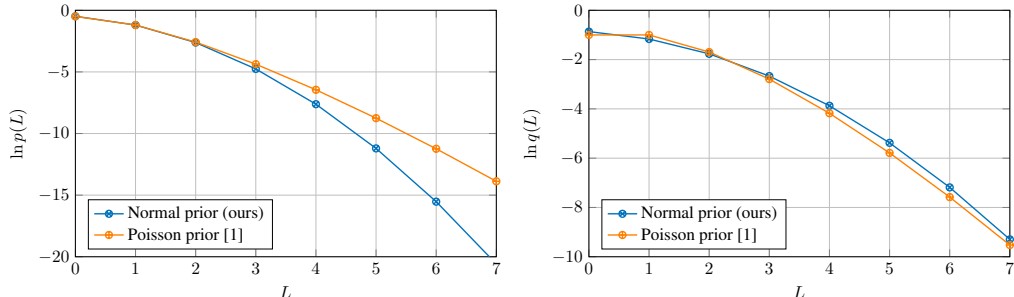

Figure 4: Log probability mass function of the (left) prior distribution over the model depth used in [1] and of the discrete truncated normal distribution used in this paper; and of the (right) initial variational posterior distributions over the model depth.

# A Experimental details

This appendix outlines the implementation details corresponding to experiments in Section 4.

## A.1 Data generation

The spiral dataset used in the experiments of Section 4 and visualized in Figure 2 are generated according to the following sampling procedure[2]:

$$t_n \sim \text{Uniform}([0, 1]) \tag{11a}$$

$$u_n = \sqrt{t_n} \tag{11b}$$

$$y_n \sim \text{Uniform}(\{-1, 1\}) \tag{11c}$$

$$x_n \sim \mathcal{N}\left(\begin{bmatrix} y_n u_n \cos\left(\omega u_n \frac{\pi}{2}\right) \\ y_n u_n \sin\left(\omega u_n \frac{\pi}{2}\right) \end{bmatrix}, 4 \cdot 10^{-4} \mathrm{I}_2\right) \tag{11d}$$

## A.2 Prior selection

For selecting the prior and initial variational posterior distributions in the experiments of Section 4, we manually align the discrete truncated distribution with the Poisson distributions. Figure 4 shows a comparison of the log probability mass function of both functions as comparisons. Most important are the segments with a high log-probability, where the priors align relatively well from visual inspection. It should be noted that some discrepancies are inevitable, but at the same time negligible as these distributions only serve as a starting point and can be optimized over.

## A.3 Training procedure

Below we describe the training procedure. Here we tried to stay as close to the experimental design of [1] as possible.

For each run we set a random seed equal to the run index. We then independently sample a train, validation and test set consisting of 1024 samples each for $\omega = 0, 1, \ldots, 30$ according to Appendix A.1. We use the following hyperparameters

- Prior on the model depth: $p(L) = \text{Poisson}(L \,|\, 0.5)$ or $p(L) = \int_L^{L+1} \mathcal{N}_{\geq 0}(l \,|\, 0, 1.15^2)\, \mathrm{d}l$.

- Initialization of variational posteriors: $q(L) = \text{Poisson}^{[0, 0.95]}(L \,|\, 1)$ or $q(L) = \int_L^{L+1} \mathcal{N}_{\geq 0}^{[0.025, 0.975]}(l \,|\, 0, 1.8^2)\, \mathrm{d}l$.

- Optimizer: Adam [12] with default hyperparameters ($\beta_1 = 0.9$ and $\beta_2 = 0.999$).

---

[2]The variance in the last step seems to differ from the original description in [1, Appendix B.1], however, the value reported there (0.02) refers to the standard deviation of the normal distribution, as verified with their publicly available experiments.

- Learning rate: $0.005$ ($0.0005$ for $\hat{\mu}_L$ and $\hat{\sigma}_L^2$, and for the rate parameter of the posterior Poisson distribution $\hat{\lambda}_L$).
- Number of epochs: 20.000.
- Batch size: 256 (randomly shuffled per epoch).
- Leaky ReLU: $\max(\alpha x, x)$, where $\alpha = 0.1$.
- Reparameterization: strictly positive parameters are transformed using the $\mathrm{softplus}$ function for unconstrained optimization.

