# OpenReview forum: "Improved Depth Estimation of Bayesian Neural Networks"
_NeurIPS.cc/2024/Workshop/BDU — NeurIPS BDU Workshop 2024 Poster_

### Official Review · Reviewer_85uS · 2024-09-16
**Overall good for workshop.**

**Rating:** 7
**Confidence:** 4

**Review:**

Quality:

The paper presents a novel approach to depth estimation in Bayesian neural networks by introducing a discrete truncated normal distribution. This method allows for independent learning of the mean and variance of the network depth, which is a significant improvement over previous methods .
The authors utilize variational inference to infer posterior distributions, effectively balancing model complexity and accuracy through the minimization of variational free energy. This indicates a solid methodological foundation.


Clarity:

The writing is generally clear, with a structured approach to presenting the problem, methodology, and results. The introduction effectively sets the stage by referencing earlier work and highlighting the improvements made.
However, some areas could benefit from further elaboration, particularly in explaining the implications of the results and the significance of the variational free energy in practical applications.

Originality:

The introduction of a discrete truncated normal distribution for modeling depth is a unique contribution that distinguishes this work from existing literature. This originality is crucial in advancing the field of Bayesian neural networks.
The paper builds on the foundational work of Nazareth and Blei, but it extends their ideas in a meaningful way, showcasing the authors' innovative thinking.

Significance:

The improvements in test accuracy on the spiral dataset and the reduction in variance of posterior depth estimates are significant outcomes that demonstrate the practical applicability of the proposed method.
The findings could have broader implications for various applications in machine learning where depth estimation is critical, thus enhancing the relevance of this research.

pros:

Innovative Approach: The use of a discrete truncated normal distribution is a novel contribution to the field.

Improved Accuracy: The method shows improved test accuracy and reduced variance in posterior estimates, which are critical for practical applications.

Solid Methodology: The use of variational inference and the Adam optimizer for training indicates a robust methodological framework.


cons:

Clarity Issues: Some sections could be clearer, particularly regarding the implications of the results and the methodology.

Limited Scope: The experiments are primarily focused on a specific dataset (spiral dataset), which may limit the generalizability of the findings.

In summary, this work presents a significant advancement in the field of Bayesian neural networks, with a clear methodology and promising results, although it could benefit from improved clarity and broader experimental validation.

---

### Official Review · Reviewer_fQUJ · 2024-09-29
**The paper proposes a novel approach for estimating the depth of BNNs, which improves upon existing methods.**

**Rating:** 7
**Confidence:** 3

**Review:**

The paper presents a novel approach for estimating the depth of Bayesian Neural Networks (BNN) by using a discrete truncated normal distribution instead of a Poisson distribution. The proposed method reduces the variance in posterior estimates of the appropriate number of layers and improves test accuracy compared to existing methods. The paper also discusses the probabilistic model and variational inference procedure, which are essential for understanding the method's effectiveness.

However, the paper has few weaknesses. First, the authors do not provide a thorough analysis of the proposed method's computational efficiency. Second, the authors only evaluated the method on a single dataset (spiral dataset), which may not be sufficient to demonstrate its generalizability. Third, the authors do not provide a detailed discussion of the method's limitations.

---

### Decision · Program_Chairs · 2024-10-09

Accept (Poster)